# Development of a potential vaccine against Capripox virus implementing reverse vaccinology and pan-genomic immunoinformatics

**Md. Mohaimenul Islam Tareq[1], Sattyajit Biswas[2], Farazi Abinash Rahman[2], Labib Sharirar Siam[1], Sadia Jannat Tauhida[1], Shamim Ahmed[1], Hasan Jafre Shovon[1], Mariya Ahmed[2], Kazi Afrin Jerin[3], Md. Nazmul Hasan[1]\***

**1** Laboratory of Pharmaceutical Biotechnology and Bioinformatics, Department of Genetic Engineering and Biotechnology, Jashore University of Science and Technology, Jashore, Bangladesh, **2** Department of Genetic Engineering and Biotechnology, Jashore University of Science and Technology, Jashore, Bangladesh, **3** Bangabandhu Sheikh Mujibur Rahman Maritime University, Dhaka, Bangladesh

\* mn.hasan@just.edu.bd

## Abstract

CPXV is responsible for animal diseases affecting cattle (Lumpy Skin Disease), sheep (Sheeppox), and goats (Goatpox). During outbreaks, these diseases have huge socio-economic effects. Now, no vaccination that is effective against sheeppox, goatpox, and lumpy skin disease is available. This work used an immunoinformatic methodology to discover possible targets for vaccination against CPXV. After the 122 CPXV proteins were obtained from the Vipr database, several investigations into the proteins' virulence, antigenicity, toxicity, solubility, and IFN-g activity were carried out. Three outer membrane and extracellular proteins were selected to predict their B-cell and T-cell epitopes based on certain distinctive features. These epitopes exhibit conservation across three species, namely Sheeppox virus (SPPV), Goatpox virus (GTPV), and Lumpy skin disease virus (LSDV) of CPXV. This will provide more comprehensive immunity against diverse virus strains worldwide. Nine MHC-I, MHC-II, and B-cell epitopes were selected to generate multi-epitope vaccine constructions. These constructs were linked using AAY, GPGPG, and KK linkers and 50S ribosomal protein L7/L12 adjuvants to enhance the immunogenicity of the vaccines. Molecular modeling and structural validation enabled the production of vaccine constructs with high-quality 3D structures. CPXV (Protein A35, Protein Resolve A22, and Scaffold Protein) was selected for further analysis because of its varied immunological and physiochemical properties (Number of Amino Acids, Molecular Weight (Daltons), Theoretical pI Aliphatic index, Grand average of hydropathicity (GRAVY), Instability index GC content, and CAI value) and docking scores. The bacterial expression system showed notable gene expression for the CPXV-V5 vaccine, as shown by computational cloning analysis. Molecular dynamics (MD) simulations revealed structural stability and long-term epitope visibility, implying strong immune responses after

**Data availability statement:** All relevant data are within the manuscript and its Supporting Information files.

**Funding:** The author(s) received no specific funding for this work.

**Competing interests:** The authors have declared that no competing interests exist.

delivery. These results suggest that the developed vaccines might be quite safe and effective in practical settings, and they offer a solid foundation for further experimental verification.

## Introduction

The genus Capripoxvirus (CPXV), in the subfamily Chordopoxvirinae of the family Poxviridae, is composed of three closely related viruses: the lumpy skin disease (LSD), goatpox virus (GPV), and sheeppox virus (SPV) [1]. The virus was named after the goat, sheep, and cattle types of animals from which it was first isolated. These viruses represent the most devastating poxvirus infections affecting production animals and are the root causes of several economically significant illnesses [2]. Capripox viruses are classified as category A diseases of the OIE because of their quick dissemination and potential to cause irreversible financial damage in the livestock sector [3,4]. Furthermore, these viruses share a significant neutralizing site, making animals who have contracted one strain of the CPXV family and survived it immune to contracting any other strain. Thus, it would be beneficial to immunize cattle against LSD using CPXV vaccine strains originating from sheep and goats [5–7]. Cattle with lumpy skin disease exhibit pyrexia, widespread skin and internal pox lesions, and widespread lymphadenopathy [8].

There are endemic strains of the three viruses throughout Africa, the Middle East, Central Asia, and the Indian subcontinent [9]. LSD is prevalent in several regions of Europe, particularly the Balkans [10]. Reports of LSD invasions into Bangladesh, China, and India in 2019 [11] were also made. One of the characteristics of Capripox viruses is that they may infect their hosts directly or indirectly. Infection may spread via two primary modes: direct contact, which occurs when aerosols released by afflicted hosts come into contact with others, and indirect contact, which happens when the surrounding atmosphere or infected vectors serve as a means of transmission [12]. Yet, among the same genus, viruses can differ in their modes of transmission [13]. The viruses responsible for sheep pox (SPV) and goat pox (GPV) are typically found in oral, nasal, or ocular secretions. Animals may acquire these viruses either by direct inhalation of aerosols or through indirect exposure [14,15]. The authors state that several insects, such as *Aedes aegypti mosquitoes, ixodid ticks*, *Stomoxys calcitrans*, and *Haematopota spp*., are included in their research. Tabanids serve as the main carriers of the LSD virus by mechanical transmission [16–21]. While the morbidity rate with LSD varies from 5% to 45%, the mortality rate usually remains about 10%. On the other hand, when an epidemic happens in naïve European cattle breeds for the first time, both rates might be significantly greater (morbidity up to 100%) [22]. Extremely contagious SPPV and GTPV can result in up to 50% death and a high morbidity rate of 70–90%. Particularly vulnerable are young lambs and youngsters, whose death rate can occasionally reach 100% [23]. The severity of the clinical sickness is mostly determined by factors such as the host species, breed, age, immunological status, and production stage. Still, there may be some variation in the virulence of different strains. Animals at the prime of their productivity, which

includes high-producing European dairy cows and sheep varieties, are frequently more severely impacted. CPXVs often cause clinical illness in sheep, goats, or cattle and are comparatively host-specific. However, there are certain exemptions wherein some variations of SPPV and GTPV can invade and affect sheep and goats. Remarkably, a recent genetic analysis conclusively identified GTPV as the only factor responsible for every pandemic seen in Ethiopian sheep and goats that were examined [24]. Red serows (*Capricornis rubidus*), wild ruminants, have been infected with GTPV recently. Mizoram, India [25]. Domestic cattle and Asian water buffalo are susceptible to the lumpy skin disease virus [26], however, some strains may also infect sheep and goats.

Commercially accessible vaccines for treating CPXV infections are limited to live-attenuated vaccines. These vaccines are developed from field isolates that have been weakened via repeated passaging in cell cultures or on the chorioallantois membrane of embryonated chicken eggs. [7,27–30]. Because all CPXVs have a single main antigen that neutralizes antibodies [10]Field immunization with both homologous and heterologous vaccines is effectively accomplished ([31,32]. In recent years, a study where they compared the effectiveness of a live-attenuated Romanian SPPV vaccine to a live-attenuated LSDV vaccine (derived from the LSDV-Neethling strain) in protecting sheep against SPPV challenge infections. The research revealed that the heterologous LSDV vaccine offered only partial protection to the sheep when exposed to SPPV challenge infection. However, homologous vaccination resulted in full clinical protection of the animals. [33]. Furthermore, it has been noted that GTPV vaccinations often offer powerful protection against SPPV infection in sheep. [34]. Despite the protective immunity provided by many live-attenuated immunizations when delivered homologous, it is important to consider many downsides. [35]. Differentiation between vaccinated and infected individuals is a known issue with live vaccines due to the replication of attenuated viruses, whereas killed vaccines, already developed in regions like Germany and Italy, do not replicate and are easier to distinguish [36]. Moreover, studies have shown that serological assays and molecular diagnostics effectively differentiate killed vaccines from natural infections, alleviating concerns about using killed vaccines. [37]. Reports have been made about vaccine failure and the occurrence of severe adverse effects similar to those seen in natural infections for CPXV live-attenuated vaccines. [38]. Additionally, the absence of a strategy to differentiate [39] Between sick and vaccinated animals (DIVA) using serological methods, together with trade restrictions, results in the prohibition of these vaccines in disease-free countries. Consequently, there are substantial challenges to using vaccines for prevention.

Hence, novel treatment strategies are needed to counter rising CPXV strains. The immune system has garnered interest as a subject of research due to its crucial role in pathogenesis and the body's defense against viral infections and malignancies. The development of bioinformatics and immunoinformatics has expedited the identification of new treatment targets against various pathogenic strains. Vaccination based on multiple epitopes is an emerging preventative approach [40–43]. To design vaccines that work, it is essential to identify immunogenic antigens [44]. Effective multi-epitope vaccine formulations include contiguous B- and T-cell epitopes within each antigenic peptide sequence, therefore stimulating both cellular and humoral immune responses against specific viral infections [40]. We used reverse vaccinology and biophysical approaches to create a multi-epitope vaccine targeting LSD, SPV, and GPV infections in our research. By analyzing the protein sequence data from the latest CPXV strains, we were able to identify the crucial B- and T-cell epitopes inside certain antigenic peptides. The lead epitopes that shared common regions were included in the vaccine constructs. The effectiveness of the proposed vaccine designs was assessed using in silico cloning into a host-vector expression system, immuno-informatics, and examination of their binding capacity with immune receptor proteins.

## Materials and methods

### Proteins retrieval, re-annotation, and pan-genome analysis

A total of 122 CPXV were obtained from the NCBI Virus database (https://www.ncbi.nlm.nih.gov/labs/virus), which is a comprehensive and publicly accessible resource that provides curated genomic and proteomic data for a wide range of viruses. It ensures high-quality annotations and reliable sequence information, making it a trusted source for virology

research, and the dataset was downloaded in the FASTA file format. After screening and excluding duplicate and error files, a total of 108 sequences of LSD, SPV, and GPV were used for further analysis. PROKKA (version 1.14.6), a program for quickly annotating de novo archaeal, bacterial, and viral genomes, was used to re-annotate the complete genome sequences with an emphasis on finding coding regions [45]. Each component's GFF3 format files were analyzed using Roary (version 3.13.0) [46]. To determine whether genes are present or absent, Roary uses the statistical multiple alignment tool PRANK v.170427 [47]. There are two important choices for doing pan-genome analysis: the minimum percentage identity for BlastP sequence matches (default value is 95%) and the percentage of isolates needed to designate a core gene (default value is 99%) [48]. The core genes were checked using Virus-mPLoc to predict their subcellular localization. [49].

## T-cell and B-cell epitope prediction

Since the outer membrane and extracellular proteins of pathogenic organisms are the most excellent prospects for designing a chimeric vaccine, surface proteins of pathogenic organisms are some of the most excellent prospects for designing a chimeric vaccine. These proteins exhibit a topology that is meant to seek out immunogenic factors. There are specific protein fragments referred to as T cell epitopes derived from the inner structure of pathogenic cells. These epitopes are located on the cell surface and are viewed by MHC (Major Histocompatibility Complex) molecules. Cytotoxic T cells, also known as CD8 + T cells, recognize MHC class I molecules. In contrast, helper T cells, or CD4 + T cells, recognize MHC class II molecules that carry foreign peptides. T cell epitopes, on the other hand, are narrower and less flexible than B cell epitopes, which are often associated with flexible and exposed antigen surface regions. The Immune Epitope Database (IEDB) server (https://www.iedb.org/) was used to predict T cell and helper T cell epitopes. The prediction of MHC-I and MHC-II binding epitope was performed using the stabilized matrix method (SMM) scoring technique and the neural network tool (netMHCpan-4.1 EL epitope predictor for MHC I and netMHCIIpan-4.1 EL epitope predictor for MHC II) [50]. Epitopes that bind at the top and overlap each other, with a half-maximal inhibitory concentration (IC50) value of 200 nM and a length ranging from 12 to 20 residues, were given higher priority. The BCpred server, available at https://webs.iiitd.edu.in/raghava/bcepred/, was developed to identify B-cell epitopes. BCpred determines linear B-cell epitopes essential for triggering immune responses that prompt B cells to generate antibodies. B-cell epitopes are distinct immunogenic sites that are continuous in structure, with a defined threshold of 0.8, and all parameters were kept at their default settings [51]. Interferon-gamma (IFN-γ), an efficient cytokine primarily produced by T cells, is essential in orchestrating numerous immune responses to combat diseases and infections. As a result, the predicted T cell epitopes have to be able to induce interferon. The IFNepitope server (https://webs.iiitd.edu.in/raghava/ifnepitope/predict.php) was implemented to predict interferon-inducing epitopes out of MHC-II binding epitopes. The IFNepitope service calculates the region of antigenic protein sequences that elicit interfere-γ (IFN-γ) response. The SVM model was used to forecast the levels of IFN-γ. The IFN-gamma vs. non-IFN-gamma model was used in particular [52].

## Epitope selection and vaccine design

Both T and B cell epitopes are important aspects when it comes to constructing multiepitope vaccines because they are effective in stimulating immune responses. The multiepitope strategy includes the whole combination of T cell and B cell epitopes, depending on their prognosis, utilizing various data processing that shapes the multiple epitope vaccine that reflects many requirements of pathogen recognition. This strategy tries to broaden and specialize the immune response, providing a more robust defense against pathogens while reducing the possibility of immune evasion. These epitopes of CD8 + T cells, CD4 + T cells, and B cells identified from the IEDB server and BCpred server (http://ailab-projects2.ist.psu.edu/bcpred/predict.html) were further evaluated for possible antigenicity, allergenicity, and toxicity using Vaxijen 2.0 (https://www.ddg-pharmfac.net/vaxijen/VaxiJen/VaxiJen.html), AllerTOP v2.0 (https://www.ddg-pharmfac.net/allertop_test/), and ToxinPred (https://webs.iiitd.edu.in/raghava/toxinpred2/batch.html). The ultimate epitopes were determined

based on antigenicity, non-allergenicity, and non-toxicity of the epitopes under consideration. The epitopes were conjugated with the adjuvant to boost the immune response, and 50S ribosomal protein L7/L12 was selected as the adjuvant for vaccine development. The epitopes were linked with the adjuvant using several linkers, beginning with the EAAAK linker. The EAAAK linker can connect the adjuvant to the developed vaccine at the N terminus to increase vaccine immunogenicity. In the context of multiepitope vaccines, linkers offer judicious spacing between multiple epitopes, allowing them to safeguard structural integrity and prevent interfering with one another. The type of linker AAY and GPGPG is used to link cytotoxic T cell and helper T cell epitopes, while the KK linker is used to connect B cell epitopes, respectively, to form a multi-epitope vaccine.

## Immunological and physicochemical properties

One of the critical procedures in developing vaccines involves thoroughly assessing their physiological attributes to ensure their safety, efficacy, and compatibility with the intricate functioning of the human body. An analysis was conducted employing the ExPASy ProtParam (https://web.expasy.org/protparam/) server, a widely recognized platform, to evaluate the physiological characteristics of the vaccine design. The ExPASy server facilitated the identification of numerous vital factors about the vaccine under consideration, which were subsequently subjected to a comprehensive review. [53]. This review encompassed an assessment of the vaccine's stability and initial screening, thereby gaining a thorough understanding of its properties. To progress further, the immunogenic attributes of the vaccine, namely its toxicity, allergenicity, and antigenicity, were also scrutinized. The investigation aimed to comprehensively understand the vaccine's potential effects on the human body and its ability to elicit an immune response.

## Homology modelling, 3D structure prediction and validation

The GalaxyWEB server predicted the designed vaccines' three-dimensional (3D) structures. The 3D structures of the designed vaccine candidates were refined by applying the Galaxy Refine 2 computational tool [54]. The demonstrated 3D vaccination structures were confirmed further using the ERRAT tool and the PROCHECK [55] within the server SAVES v6.0 (https://saves.mbi.ucla.edu/) and ProSA-Web (https://prosa.services.came.sbg.ac.at/prosa.php).

## Molecular docking

Identifying the binding interactions between desired ligands and proteins is made possible by molecular docking experiments. [56]. Toll-like receptors TLR-3/8 are involved in CCHF viral infection. [57], and TLR 4 has been established as a critical agent to initiate viral pathogenesis to produce inflammation in the process [58]. Thus, we selected these TLRs as our receptors, and the vaccine model was developed as the ligand. The molecular docking study was accomplished utilizing ClusPro v2.0 web tools (https://cluspro.bu.edu/login.php) [59]. More recently, ClusPro has been demonstrated to be more reliable than other ways of docking unbound protein structures. [60]Perhaps because the final selection depends on the size of the cluster and not the score from the function [61,62]. Before the docking, the TLRs were prepared by removing water molecules using Maestro v. v-11.3. The web tools dock the protein-ligand complex by conducting compact bond docking, employing a procedure that involves clustering of the lowest energy structure and energy minimization. According to the center, with a low energy score and cluster members, the best-docked complex was selected. [63]. Maestro v-11.3 was used to visualize the structure of the vaccine complex and to analyze the interaction. [64].

## Codon adaptation and in-silico cloning

Disulfide bonds were introduced into the vaccine model to improve the stability of vaccines. Disulphide engineering, a modern and innovative method, can be implemented using the Disulphide by Design 2.12 web server. [65]. The final vaccine structure (V2) was subjected to codon optimization and silico cloning. The Java Codon Adaptation (JCat) tool was employed for codon optimization of the vaccine design, aiming to enhance the expression of the cloned sequence within

the E. coli expression system. [66]. JCat Figs out GC content and codon adaptation index (CAI) to evaluate the cloned sequence's expression. CAI value should strive to be 1, and the GC content of the modified sequence needs to range approximately 30–70%, indicating favourable transcriptional and translational efficiency [67]. Lastly, Snap Gene (https://www.snapgene.com/) was employed to clone the most influential vaccine design in the pET-28a (+) expression vector.

## Immune simulation

The immunological activity of the vaccine was evaluated using computerized immunological experiments on the C-ImmSim tool. [68]. This tool forecasts and analyzes epitope and immune-mediated contacts using a position-specific score matrix (PSSM) and multiple algorithms. The immunological simulation studies for the vaccinations followed specified criteria, such as a four-week dosage gap and time steps for the three injections set at 1, 84, and 168 over four weeks. The overall simulation measurement was initially set at ten, while the immune simulation evaluation was performed using a thousand simulation cycles.

## Molecular dynamics simulation

To study the behavior of the epitope at the molecular level and the extent of stability of such a protein-protein interaction, a molecular dynamics computational approach was employed. The vaccines can exist in the monomers and M-shape dimers, where the latter is before the activation of signal transduction. In these regards, three docked vaccine complexes (Protein A35, Protein Resolve A22, Scaffold Protein) assessed the effects of glycosylation and the thermodynamic stability of the developed epitope vaccine. To assess the thermodynamic compatibility of the protein-ligand complexes, three vaccine receptor complexes were chosen for 100 ns MD simulations. The molecular dynamic simulations, which evaluated different structures of protein-ligand complexes, were conducted using the Schrödinger "Desmond v3.6 Program" (Paid version) within a Linux framework. The given method utilizes the TIP3P aqueous model to generate a desired volume with a periodic boundary condition. This condition includes an orthorhombic form with a division of 10 Å. Relevant ions, such as 0+ and 0.15 M salt (Na+ and Cl−), have been used and uniformly dispersed during randomization inside their chemical solvent environment to neutralize electrical charge in their formation. The solvency protein structures were constructed using combinations of agonists. Subsequently, the system framework was optimized and became skilled in using the protocol that used force field constants, namely OPLS3e, which is included inside the Desmond package. The NPT assemblies were maintained at a temperature of 300 K and a pressure of 1.01325 bar using global Nose-Hoover temperature pairings and an isotropic technique. The assemblies included 50 PS grabbing halts with an efficiency of 1.2 kcal/mol. The fidelity of the molecular dynamics simulation was assessed using the Simulation Interaction Diagram (SID) from the Desmond modules of the Schrödinger Suite. The stability of this protein-ligand complex was evaluated by considering descriptors such as Molecular Surface Area (MolSA), Polar Surface Area (PSA), Solvent-Accessible Surface Area (SASA), Protein-Ligand Interactions (P-L), and Intermolecular Hydrogen Bonds.

The Schrödinger Maestro program version 9.5 was used to make pictures of Molecular Dynamics Simulations. The Simulations Interaction Diagram (SID) of the Desmond modules of the Schrödinger software was used to assess the potential simulation scenario and the accuracy of the molecular dynamics simulation. To assess the stability of the Vaccine-Receptor complex structures, we analyzed the Secondary structural element (SSE) root-mean-square deviation (RMSD) and root-mean-square fluctuation (RMSF) using the work trajectory data.

## Results

### Construction of the pan-genome and identification of core proteins

The collective pan-genome of the 108 Capripox viruses consisted of 257 genes. Among them, 112 genes were identified as core genes, since they were present in all CPXV genomes with a minimum similarity threshold of 90%. Fig 1a and 1b

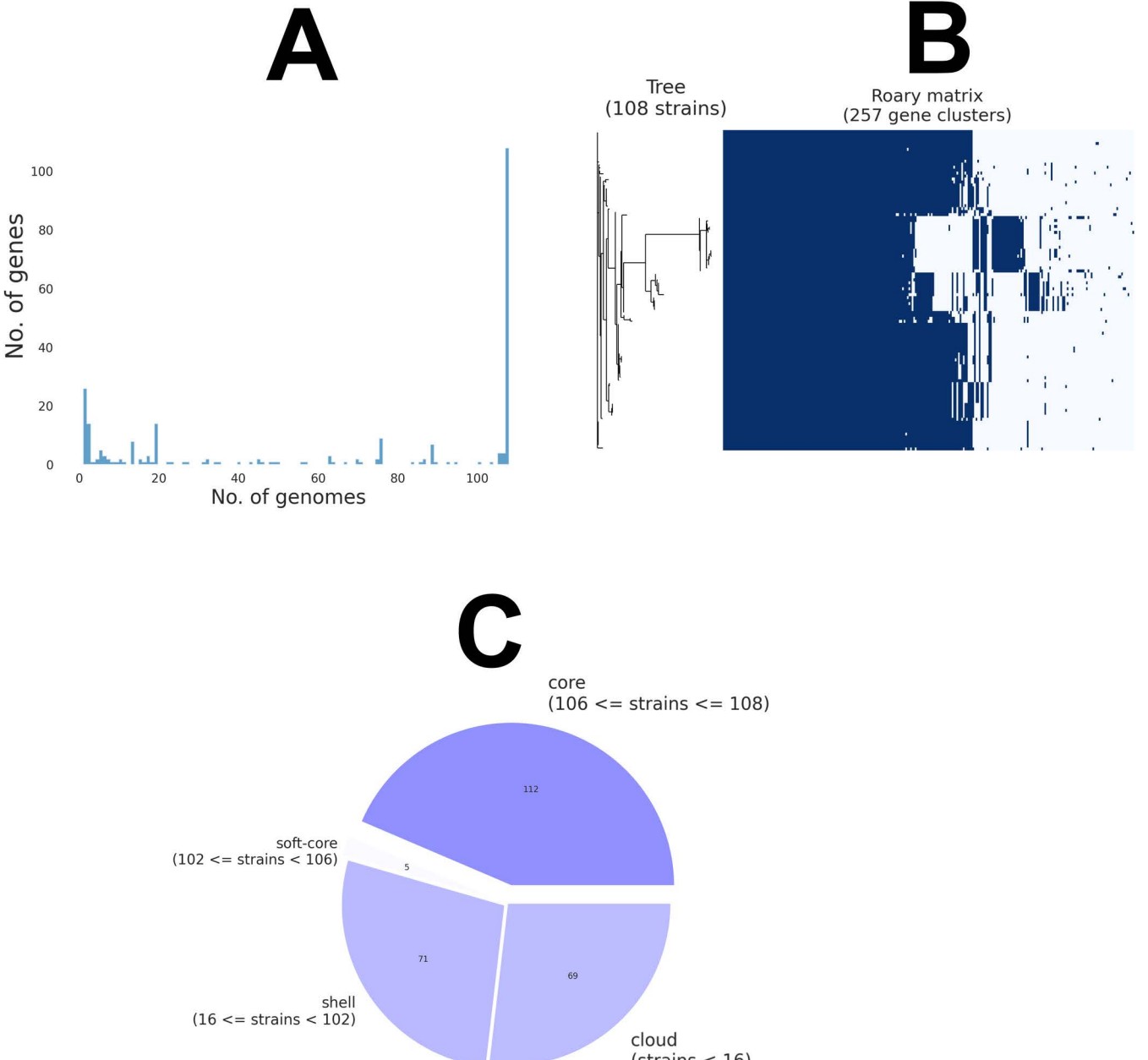

**Fig 1. Pan-genome analysis. a) The number of genomes vs the total number of genes. b) Clustering of the genes. c) A pie chart represents several core, soft, shell, and cloud genes.**

display a matrix that represents the presence and absence of core and auxiliary genes, together with a full genome phylogenetic tree. The examination of the whole pan-genome showed that there were 112 essential genes (43.58%) present in all 102 isolates, 5 genes that were commonly found in most isolates (1.94%), 71 genes that were present in a subset of isolates (27.6%), and 69 genes that were found in less than 16 samples (26.85%) Fig 1c. This indicates that the bulk of the genes belong to the core gene set. Vaccine target identification was performed using 112 core genes that met a cut-off value of 100%.

Furthermore, proteins located on the outer membrane were selected for further study by predicting their subcellular localization using Virus-mPLoc.

### T-cell and B-cell epitope prediction

The Study meticulously prioritized the top three major antigenic proteins that do not cause allergy and toxic effects. These proteins were then considered in an exhaustive analysis to ascertain lead epitopes for the construction of a chimeric vaccine against CPXV. Additionally, overlapped B-cell epitopes were found to have a BCpred score higher than 0.8 and 75% specificity. Each of the epitopes derived from these servers underwent a thorough examination for antigenicity, IFN-positivity, allergenicity, and toxicity. Based on these criteria, three overlapping lead epitopes were selected and subsequently employed in designing vaccine constructs. According to Table 1, five vaccine models were created using nine distinct epitopes to ensure the most influential vaccine. The aim of the design in the case of a multi-epitope vaccine is to induce a broader and stronger immune response against a pathogen by simultaneously targeting multiple antigenic epitopes. This approach is meant to provoke a broader and deeper reaction on the part of the immune system, as well as raising the chances of provoking various components of the immune system and encouraging the formation of different types of immune cells, including antibodies and T cells. The identified epitopes were shown to be conserved across different CPXV strains. Employing conserved epitopes in a multi-epitope vaccination promotes greater defense against various CPXV strains. Table 2 indicates the conservation of assigned epitopes.

### Vaccine construction

To construct a multi-epitope vaccine, Linkers were used to amalgamate the selected epitopes. AAY, GPGPG, and KK linkers were utilised to join five CTL epitopes, five HTL epitopes, and four B-cell epitopes. Additionally, the EAAAK linker was adjuvanted with profilin at the N-terminus to create a single construct, utilizing a comprehensive response. Finally, a multi-epitope vaccination candidate including 370 amino acid residues was eventually developed Fig 2.

### 3D structure prediction and validation

The 3D vaccine structure must be persistent and effective in examining the molecular relationships involving the host's immunological receptor protein. The Ramachandran plots evaluated the quality of vaccine constructs to validate the 3D structure. The Ramachandran plots revealed that the favored regions appeared 92.4%, 90.8%, 93.1%, 89.7%, and 90.5% for V1, V2, V3, V4, and V5 (Fig 3). Similarly, the overall quality score and Z score (Fig 4) of refined vaccine constructs were calculated by ERRAT and ProSA-webserver, with a range of values. Overall findings from the ERRAT online tool, & the ProSA servers demonstrated a superior quality of the proposed vaccine 3D structures.

### Molecular docking

An assessment of molecular docking revealed how well the vaccine structures were bound to the receptor. Table 3 presents the various configurations of the dock score. The chosen docked complex was determined based on three specific criteria: the central location inside the active site, the low energy score, and the inclusion of cluster members. The TLR-4 complex proteins (TLR4- Resolve Protein A22, TLR4- Protein A35, TLR4- Protein J5, TLR4- Protein L1, and TLR4- Scaffold Protein D13) exhibited a binding score of − 888.4, −1261.1, −1295.1, −1063.7, and −1476.7 at the center. The lowest energy was observed as − 1192.6, −1505.8, −1455.9, −1264.3, and −1476.7 respectively. These vaccine complexes exhibited high affinity and robust interaction. Furthermore, the complex files are also shown in Fig 5.

### Immune simulation

The subsequent effects elicited through the targeted vaccine design increased significantly, as estimated by immune stimulation. In general, this tendency complies with actual immune response generation. The enhanced secondary

**Table 1. Epitope prediction for the target protein.**

| Protein IDs | MHC-I Epitopes | Score | MHC-II Epitopes | Score | IFN-Gamma Positive Score | B-Cell Epitopes | BCPred Score | Allergenicity-AllerTOP2. | Toxicity |
|---|---|---|---|---|---|---|---|---|---|
| Protein L1 | TGTTTNFEF | 1.0169 | PYNVVSLNIHPFPN | 0.30286996 | 0.083365912 | NFEFINSGTSQGICAI | 0.67 | Non allergen | Non-Toxin |
| | SLTPDQKAY | 1.2139 | LIDEKEQYFYLGTAYD | 0.3804596 | 0.43572597 | CAAPTGTTTNFEFINS | 0.77 | | |
| | LSMVFLYYV | 1.2005 | IDEKEQYFYLGTAYDIV | 0.30475693 | 0.22198471 | KLEQTAEATAEAKCDI | 0.76 | | |
| | APTGTTTNF | 1.6244 | IDEKEQYFYLGTAYDIVN | 0.39685701 | 0.28281613 | AVVVILSMVFLYYVKK | 0.74 | | |
| | YQFYIIAAV | 1.1395 | PYNVVSLNIHPFPNNF | 0.30092346 | 0.21801638 | DVTTKASTKFSPSQSS | 0.87 | | |
| Protein J5 | NIKIKNGYI | 0.8536 | IIKIGEETRLPYY | 0.7904 | 1 | GEETRLPYYCWYEPCK | 0.82 | Non allergen | Non-Toxin |
| | HPIWLPISL | 1.5651 | SIIKIGEETRLPYY | 0.709 | 1 | KKNISLCNISDCRVTL | 0.79 | | |
| | KRSDALIVK | 1.5128 | | | | DALIVKSLKKNISLCN | 0.79 | | |
| | | | IGEETRLPY | 1.073 | | RCRCLNPDSSIIKIGE | | 0.65 | |
| | | | WLPISLFII | 0.8517 | | GEETRLPYYCWYEPCK | | 0.82 | |
| ProteinA35 | DEKEQYFYL | 1.3471 | TLGPYNVVSLNIHPFP | 1.1376 | 0.14581055 | VSLNIHPFPNNFIEQS | 0.7 | Non allergen | Non-Toxin |
| | YTLITTLGV | 1.1059 | LGPYNVVSLNIHPFPN | 1.1171 | 0.28453266 | MCLTDKSGWCIVDIKN | 0.7 | | |
| | QYFYLGTAY | 0.8199 | TLGPYNVVSLNIHPFPNN | 1.0397 | 0.81577613 | LGVLKIKKEISKVCS | 0.69 | | |
| | SNNSFIFSF | 0.9173 | LGPYNVVSLNIHPFPNN | 1.0149 | 0.47142637 | YTLITTLGVLKIKKE | 0.7 | | |
| | YTLITTLGV | 1.1059 | PYNVVSLNIHPFPN | 1.0259 | 0.083365912 | | | | |
| Resolvase A22 | PYIKFIHFI | 1.7026 | PARTVLEIESNNIK | 0.5877 | 0.35985329 | GKKIDDVADSFNIALR | 0.68 | Non allergen | Non-Toxin |
| | KSTESFLNW | 1.9791 | NPARTVLEIESNNIK | 0.5428 | 0.38296976 | PVMVGYSYKDRKKKST | 0.93 | | |
| | IKKEVICAF | 1.2708 | PARTVLEIESNNIKII | 0.5736 | 0.79075756 | LERQSKRSPYIKFIHF | 0.66 | | |
| | FNIALRFVL | 1.1899 | NPARTVLEIESNNIKI | 0.5827 | 0.54025271 | HIKKEVICAFDIGAKN | 0.85 | | |
| | IKKEVICAF | 1.2708 | | | | | | | |
| Scaffold protein D13 | FEIRDQYIT | 0.795 | TTLGVLKIKKEISKV | 0.6866 | 0.7112244 | CVLGTRTLKFNFTPHT | 0.87 | Nonallergen | Non-Toxin |
| | MEVRFGNDV | 1.351 | TLGVLKIKKEISKVC | 0.5512 | 0.8901613 | KFGYVTYVGYKSIQHV | 0.79 | | |
| | SRMEVRFGN | 0.9209 | PYNVVSLNIHPFPNNFI | 0.9075 | 0.74281414 | IVNIQDVDVFIKIDNV | 0.87 | | |
| | ESATIYYYI | 0.8726 | EKEQYFYLGTAYDIVNSN | 0.8999 | 0.33458311 | TARGKDKLSVRVIFSS | 0.69 | | |
| | FNFTPHTFF | 0.8834 | LIDEKEQYFYLGTAYDI | 0.8504 | 0.54775491 | KTDIISRMEVRFGNDV | 0.83 | | |
| | FIYVTELSF | 1.2164 | | | | | | | |

The top-prioritized MHC-I and MHC-II T-cell epitopes and B-cell epitopes and their corresponding immunogenic properties.

**Table 2. Comparative conservation analysis.**

| Protein IDs | MHC-I Epitopes | Conservation | MHC-II Epitopes | Conservation | B-Cell Epitopes | Conservation |
|---|---|---|---|---|---|---|
| Protein L1 | TGTTTNFEF | 0.65% (1/155) | PYNVVSLNIHPFPN | 0.65% (1/155) | NFEFINSGTSQGICAI | 0.65% (1/155) |
| | SLTPDQKAY | 0.65% (1/155) | LIDEKEQYFYLGTAYD | 0.65% (1/155) | CAAPTGTTTNFEFINS | 0.65% (1/155) |
| | LSMVFLYYV | 0.65% (1/155) | IDEKEQYFYLGTAYDIV | 0.65% (1/155) | KLEQTAEATAEAKCDI | 0.65% (1/155) |
| | APTGTTTNF | 0.65% (1/155) | IDEKEQYFYLGTAYDIVN | 0.65% (1/155) | AVVVILSMVFLYYVKK | 0.65% (1/155) |
| | YQFYIIAAV | 0.65% (1/155) | PYNVVSLNIHPFPNNF | 0.65% (1/155) | DVTTKASTKFSPSQSS | 0.65% (1/155) |
| Protein J5 | NIKIKNGYI | 0.65% (1/155) | IIKIGEETRLPYY | 0.65% (1/155) | GEETRLPYYCWYEPCK | 0.65% (1/155) |
| | HPIWLPISL | 0.65% (1/155) | SIIKIGEETRLPYY | 0.65% (1/155) | KKNISLCNISDCRVTL | 0.65% (1/155) |
| | KRSDALIVK | 0.65% (1/155) | | | DALIVKSLKKNISLCN | |
| | IGEETRLPY | 0.65% (1/155) | | | RCRCLNPDSSIIKIGE | |
| | WLPISLFII | 0.65% (1/155) | | | | |
| Scaffold Protein D13 | FEIRDQYIT | 0.65% (1/155) | TTLGVLKIKKKEISKV | 0.65% (1/155) | CVLGTRTLKFNFTPHT | 0.65% (1/155) |
| | MEVRFGNDV | 0.65% (1/155) | TLGVLKIKKKEISKVC | 0.65% (1/155) | KFGYVTYVGYKSIQHV | 0.65% (1/155) |
| | SRMEVRFGN | 0.65% (1/155) | PYNVVSLNIHPFPNNFI | 0.65% (1/155) | IVNIQDVDVFIKIDNV | 0.65% (1/155) |
| | ESATIYYYI | 0.65% (1/155) | EKEQYFYLGTAYDIVNSN | 0.65% (1/155) | TARGKDKLSVRVIFSS | 0.65% (1/155) |
| | FNFTPHTFF | 0.65% (1/155) | LIDEKEQYFYLGTAYDI | 0.65% (1/155) | KTDIISRMEVRFGNDV | 0.65% (1/155) |
| | FIYVTELSF | 0.65% (1/155) | | | | |
| Resolvase A22 | PYIKFIHFI | 0.65% (1/155) | PARTVLEIESNNIK | 0.65% (1/155) | GKKIDDVADSFNIALR | 0.65% (1/155) |
| | KSTESFLNW | 0.65% (1/155) | NPARTVLEIESNNIK | 0.65% (1/155) | PVMVGYSYKDRKKKST | 0.65% (1/155) |
| | IKKEVICAF | 0.65% (1/155) | PARTVLEIESNNIKII | 0.65% (1/155) | LERQSKRSPYIKFIHF | 0.65% (1/155) |
| | IKKEVICAF | 0.65% (1/155) | NPARTVLEIESNNIKI | 0.65% (1/155) | HIKKEVICAFDIGAKN | 0.65% (1/155) |
| | FNIALRFVL | 0.65% (1/155) | | | | |
| Protein A35 | DEKEQYFYL | 0.65% (1/155) | TLGPYNVVSLNIHPFP | 0.65% (1/155) | VSLNIHPFPNNFIEQS | 0.65% (1/155) |
| | YTLITTLGV | 0.65% (1/155) | LGPYNVVSLNIHPFPN | 0.65% (1/155) | MCLTDKSGWCIVDIKN | 0.65% (1/155) |
| | QYFYLGTAY | 0.65% (1/155) | TLGPYNVVSLNIHPFPNN | 0.65% (1/155) | LGVLKIKKKEISKVCS | 0.65% (1/155) |
| | SNNSFIFSF | 0.65% (1/155) | LGPYNVVSLNIHPFPNN | 0.65% (1/155) | YTLITTLGVLKIKKKE | 0.65% (1/155) |
| | YTLITTLGV | 0.65% (1/155) | PYNVVSLNIHPFPN | 0.65% (1/155) | | |

The epitope conservancy of the chosen B- and T-cell epitopes has been determined using IEDB.

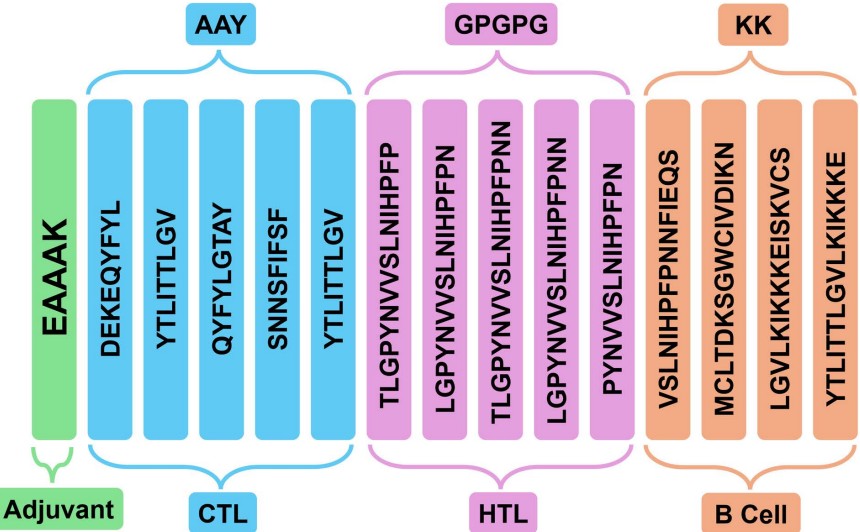

**Fig 2. Vaccine construction.** The construction of the CPXV-V5 vaccine of Protein A35, termed TLR4 Adjuvant-vaccine (V5).

responses triggered by the prioritized vaccine design increased significantly, nearly matching immune modeling expectations. The pattern seen has been ascribed to the onset of prompt immune system responses. The predominant simulated response was determined by elevated IgM levels. The secondary and tertiary immune responses exhibited significant enhancement in B cell levels and increased numbers of IgG1 + IgG2, IgM, and IgM + IgG antibodies. Nevertheless, there was a decrease in the total levels of antigen concentrations. Similarly, there were increased responses seen in cytotoxic T cells, T helper cells, macrophages, dendritic cells, natural killer cell populations, IFN-y, and interleukins. The results demonstrate that the intended vaccine design generated effective immunity against Capripox Viruses Fig 6.

## Molecular dynamic simulation

Molecular dynamics simulation (MDS) allows for the assessment of the conformational stability of molecules and atoms by simulating the system at the atomic level. An important advantage and advancement of MD simulation is its ability to illustrate the stability of a ligand inside a specific protein macromolecule. A 100 nanosecond molecular dynamics (MD) simulation was used to examine the structural organization of the chosen vaccine complex. The purpose of this action was to assess the ligands' potential to interact with the protein, namely the cavity located in its active site. The study of the molecular dynamics (MD) simulation has been elucidated using RMSF, RMSD, and SSE.

**RMSD.** The concept of "root means square deviations," or RMSD, explains how stable the vaccine complex is. The three vaccine complexes in this 100-nanosecond MDS produce good results. The vaccine complex scaffold protein with the most significant deviation among the three was found. 53.149 is the most considerable variation, occurring at 84.4 nanoseconds in Fig 7 and S1 Fig. Protein Resolve A22 performs better than scaffold protein; its highest deviation rate is 33.515, occurring at a time of 67 nanoseconds. Overall, the outcomes are favourable. However, Protein A35 yields the best outcomes and fluctuates at a far lower rate than the other. The maximum variance, or 29.873, happens at 50.8 nanoseconds.

**RMSF.** Root mean square fluctuations, often known as RMSF, provide an extra indication of stability. The MDS results indicate that the scaffold protein exhibits the most variability among the three proteins. According to Fig 8 and S2 Fig, the

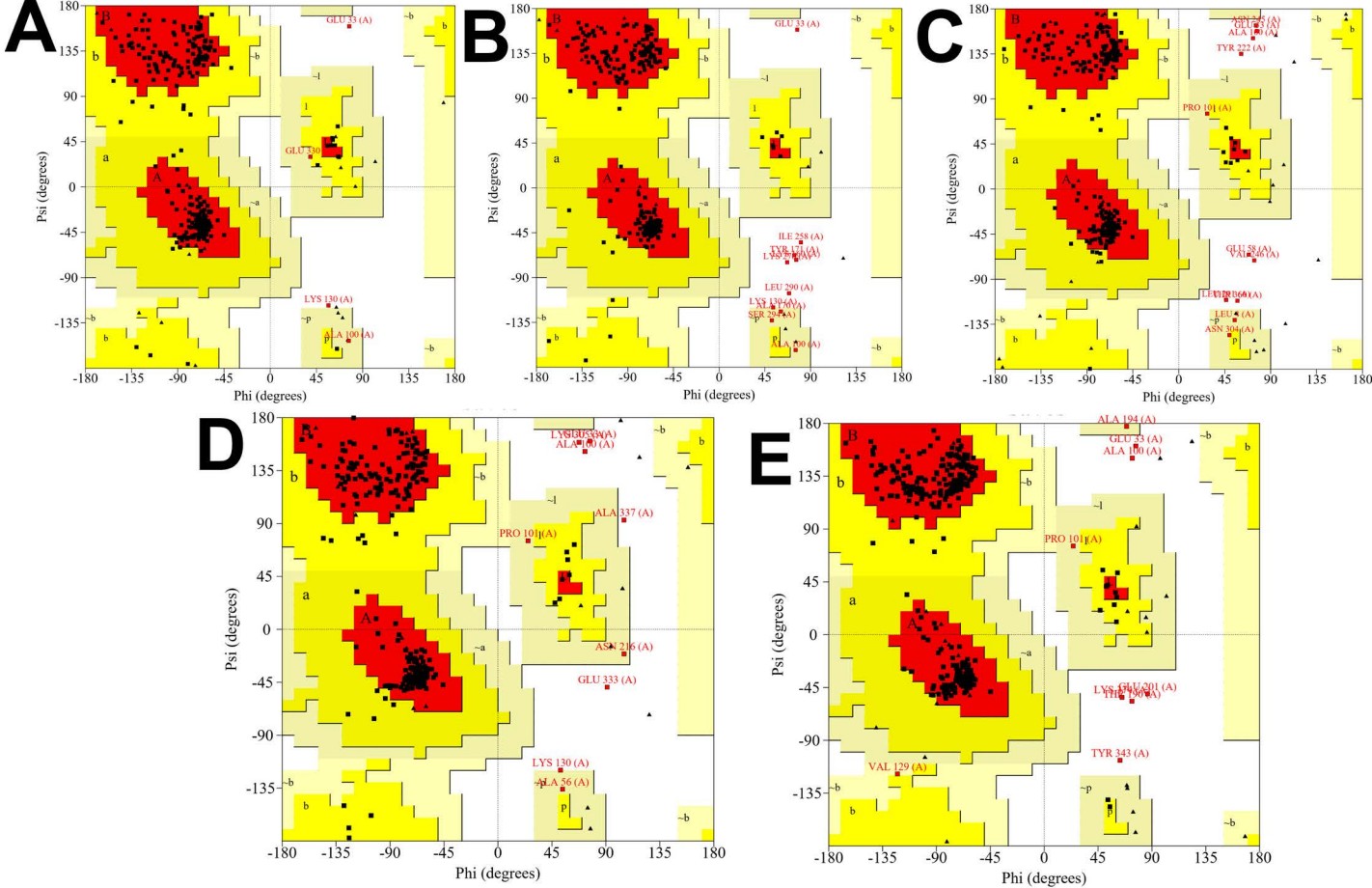

**Fig 3. Evaluation of quality assessment for CPXV vaccines.** The Ramachandran plot assessment displays **(A)**. For CPVX-V1, 92.4% of the residues are located within favored regions, 6.3% in the allowed region, and 0.6% in the disallowed region. **(B)** For CPVX-V2, 90.8% of the residues are located within favored regions, 5.3% in the allowed region, and 1.5% in the disallowed region. **(C)** For CPVX-V5, 90.5% of the residues are located within favored regions, 6.2% in the allowed region, and 0.7% in the disallowed region. **(D)** For CPVX-V4, 89.7% of the residues are located within favored regions, 7.7% in the allowed region, and 0.7% in the disallowed region. **(E)** For CPVX-V3, 93.1% of the residues are located within favored regions, 4.6% in the allowed region, and 0.6% in the disallowed region.

RMSF values for this protein are 31.015 and 8.601 at the minimum and maximum, respectively. Nearly identical outcomes are seen by a different protein known as Protein Resolve A22. In this instance, the greatest and minimum RMSF values are 5.252 and 21.674, respectively. The final protein, Protein A35, exhibits excellent outcomes compared to the other two. The graph displays the RMSF's minimum and highest values, 3.517 and 21.991, respectively.

**SSE.** Changes in a protein's secondary structure throughout simulation must be analyzed to investigate the stability of protein conformation. Fig 9 illustrates the role of certain amino acid residues, namely β-helices (shown by blue bars) and β-sheets (represented by brown bars), in the formation of secondary structures. It is evident that the majority of the amino acid residues in domain I and domain II, which constitute α-helices and β-sheets in the original structure, consistently maintain Mpro's secondary structural conformation throughout the simulation, accounting for approximately 100% of the simulation duration. Comparably, domain III's amino acid residues—which in the native structure form β-sheets—continue to maintain the protein's conformation for almost all of the simulation period Fig 9.

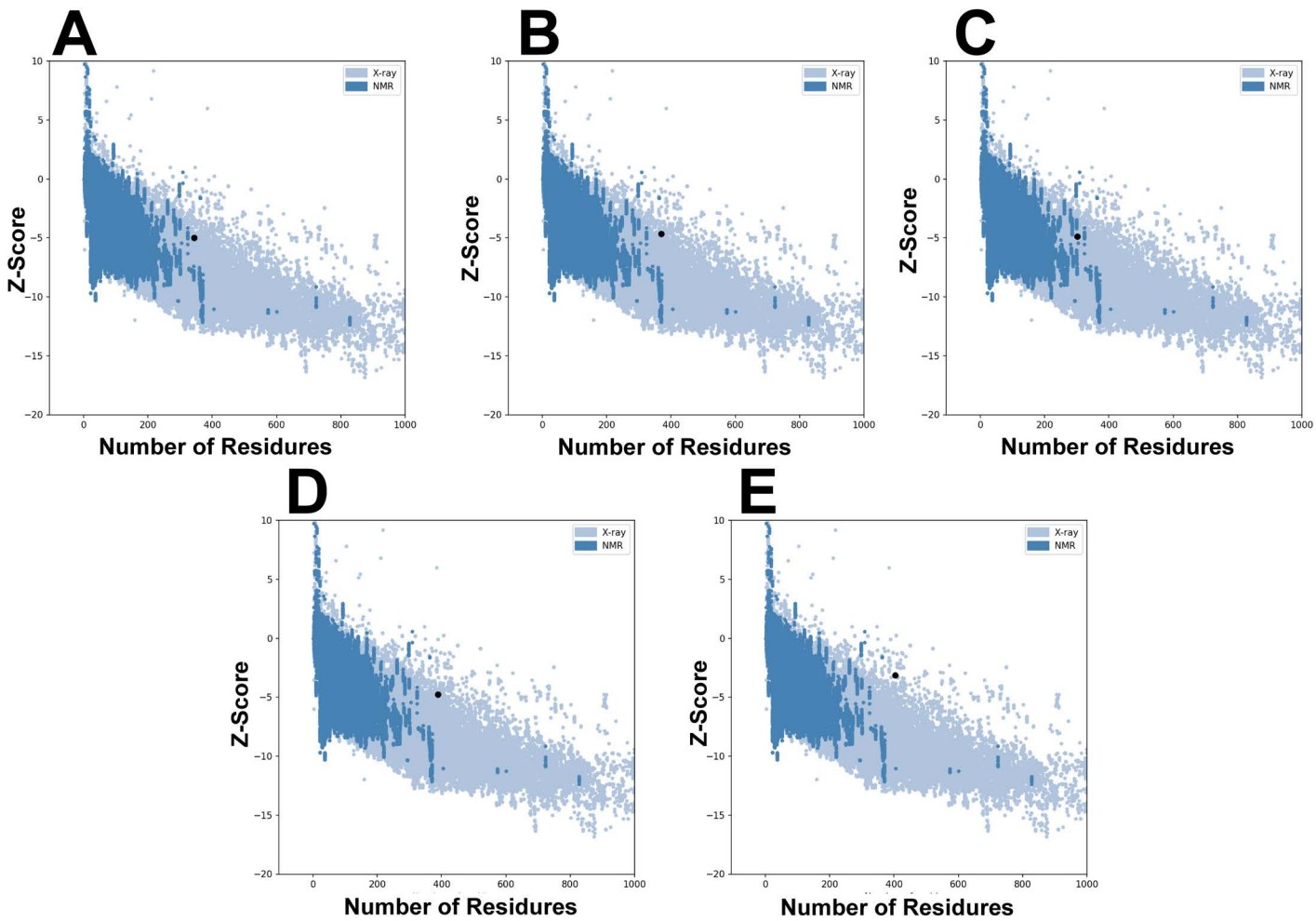

**Fig 4. 3D structure validation.** According to the ProSA-web findings, the analysis of quality validation for CPXV vaccines shows a Z-score of **(A)** −2.3 for CPVX-V1. **(B)** −2.86 for CPVX-V2. **(C)** −4.79 for CPVX-V5. **(D)** −2.63 for CPVX-V4. **(E)** −4.79 for CPVX-V3.

**Table 3. Docking score analysis of ligand-receptor complexes.**

| Serial Number | Protein Name | Docking Score | |
|---|---|---|---|
| 1 | Resolve Protein A22 | Center | −888.4 |
| | | Lowest Energy | −1192.6 |
| 2 | Protein A35 | Center | −1261.1 |
| | | Lowest Energy | −1505.8 |
| 3 | Protein J5 | Center | −1295.1 |
| | | Lowest Energy | −1455.9 |
| 4 | Protein L1 | Center | −1063.7 |
| | | Lowest Energy | −1264.3 |
| 5 | Scaffold Protein D13 | Center | −1476.7 |
| | | Lowest Energy | −1476.7 |

Determining docking scores between several vaccines of CPXV and TLR4 receptor complex with proteins.

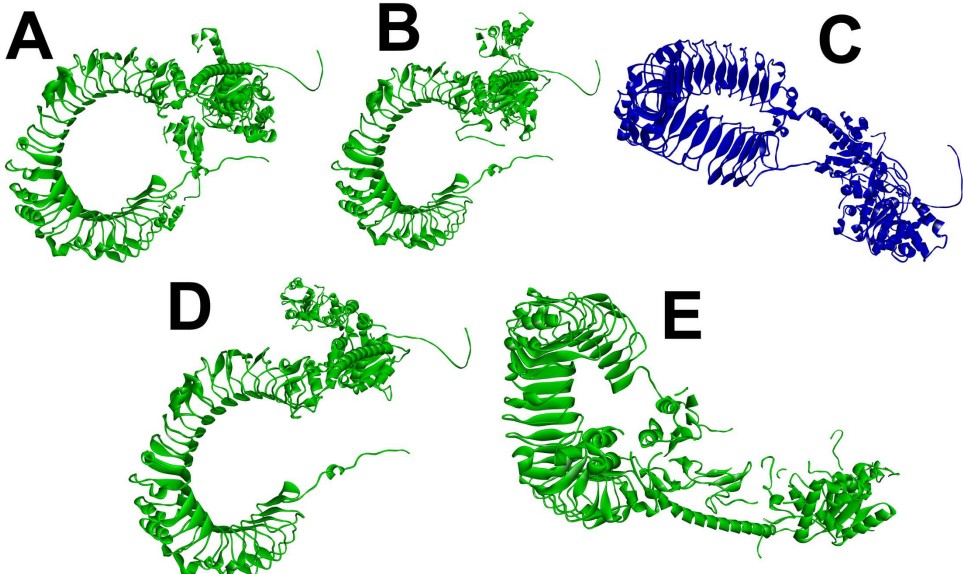

**Fig 5. Illustrating the molecular interaction between CPXV vaccines and the host immune receptor TLR4.** Herein, the interactions between **(A)** the Protein L1 vaccines and TLR4 receptors are demonstrated. **(B)** the Protein J5 vaccines and TLR4 receptor. **(C)** the Protein A35 vaccines with TLR4 receptor. **(D)** the Resolve Protein A22 with immune receptor TLR4. **(E)** the Scaffold Protein D13 vaccines with TLR4 receptor.

## Codon optimization and *In-Silico* cloning

The vaccine construct's expression in the *E. coli* expression system was analysed using the JCat web-based tool to investigate Codon Adaptation. The CAI values were discovered to be more than 0.92, and the projected GC content was found to be above 45%"in the Table 4. These findings indicate that the degree of increased vaccine expression in the E. coli strain K12 is adequate. Ultimately, by using Snap Gene software, an improved sequence, and the final vaccine design, V5 was successfully inserted into the pET28a (+) vector plasmid in Fig 10.

## Discussion

Our research is focused on novel next generation approaches to accelerating the development of vaccines. Some of the methods included are reverse vaccinology, subtractive proteomics, immunoinformatic, and vaccine informatics. The stated approaches offer a range of database servers and tools for identifying the most suitable pathogen protein for vaccine development. Subsequently, using methods based on data concerning immune responses to epitopes, highly antigenic, non-allergic, non-toxic, and safe potential epitopes can be predicted and be used directly in experimental analysis to test the vaccine.

Presently, multiepitope-based vaccines targeted at viral diseases created by various computational approaches can be used. There are many advantages to the use of computational methods in designing multiepitope-based vaccines. The techniques can skip the entire genome to obtain the DNA stretch that has the potential to elicit the strongest immune response. Moreover, it is considered that the epitopes revealed by the methods have not had increases in viral pathogenicity. The objective of this study is to identify the source of suitable epitopes from the CPXV protein by conducting a pan-genome analysis. The aim is to develop vaccines that contain multiple epitopes to control viral diseases effectively.

To predict the potential virus epitopes, virus proteins were downloaded via a pan-genomic approach to check their antigen characteristics with the help of various tools. Finally, five proteins, namely Protein L1, Protein J5, Scaffold protein D13, Resolvase A22, Protein A35 were selected for separate vaccination.

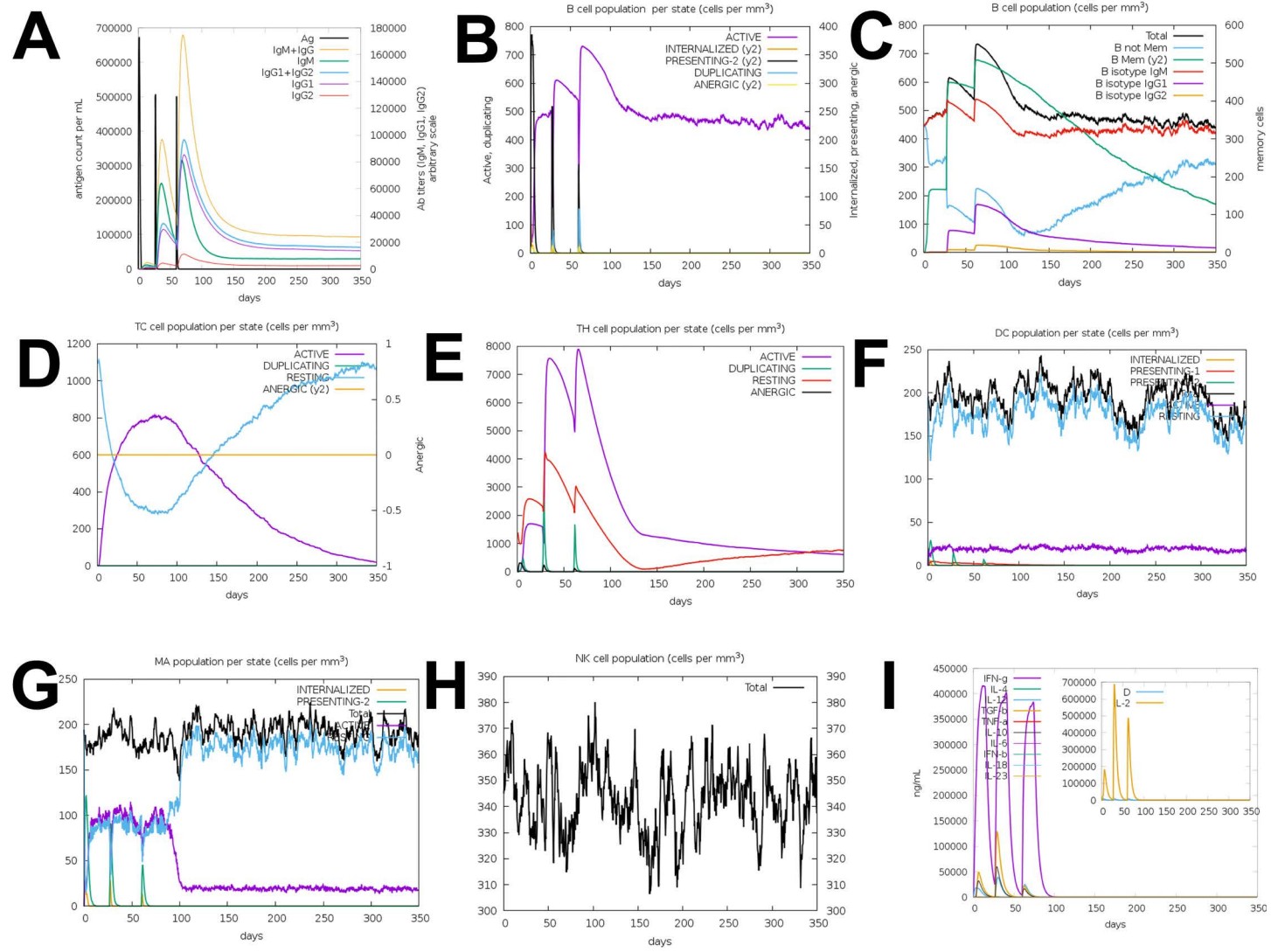

**Fig 6. In silico production of an immune reaction using vaccine as antigen. (A)** Production of immunoglobulins and B-cell isotypes upon antigen exposure; **(B)** number of engaged B-cell numbers per state; **(C)** the number of plasma B-lymphocytes and their isotypes per state; **(D)** helper T-cell population; **(E)** helper T-cell population condition throughout following immune reactions; **(F)** amount of cytotoxic T-cells per antigen-exposed state; **(G)** macrophage population activity in three subsequent immune responses; **(H)** Dendritic cell population per state and **(I)** Cytokine and interleukins production with Simpson index of immune response.

MHC-I and MHC-II T-cell epitopes are viral for using adaptative immunity information. Whereas the MHC-II epitopes can generate the cellular as well as the humoral immune responses, MHC-I epitopes can develop long-term immunity to boot out the viruses and the infected cells from the host [69]. These epitopes lead to the activation of CD4+helper T cells, which in turn leads to the production of CD8+T cell memory cells and B-cell activation [70,71]. MHC-I and MHC-II epitopes that overlapped B-cell epitopes were chosen and coupled utilizing four distinct adjuvant peptide sequences and several combinations of immune enhancers to build vaccine constructions. The multiepitope constructions created had strong antigenicity ratings according to Vaxijen v2.0. Every vaccine construct made did not contain any toxins or allergens. All these immunological features enhance its chances of being a good vaccine candidate.

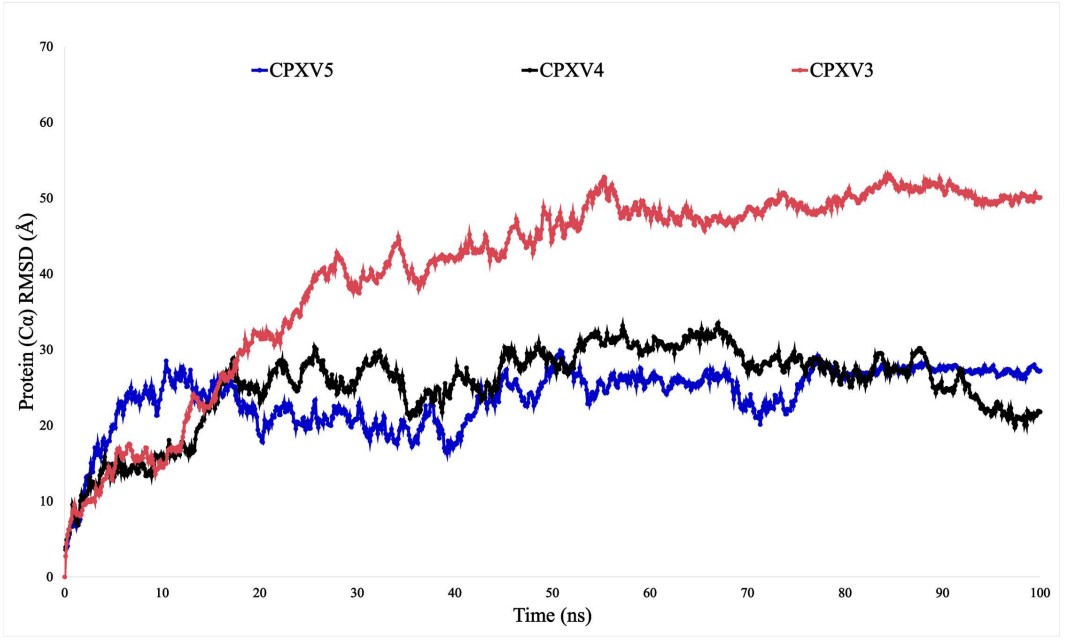

**Fig 7. RMSD analysis from molecular dynamic (MD) simulation.** Protein A35, Protein Resolve A22, and Scaffold Protein, which are blue, black, and red, serve as examples of RMSD.

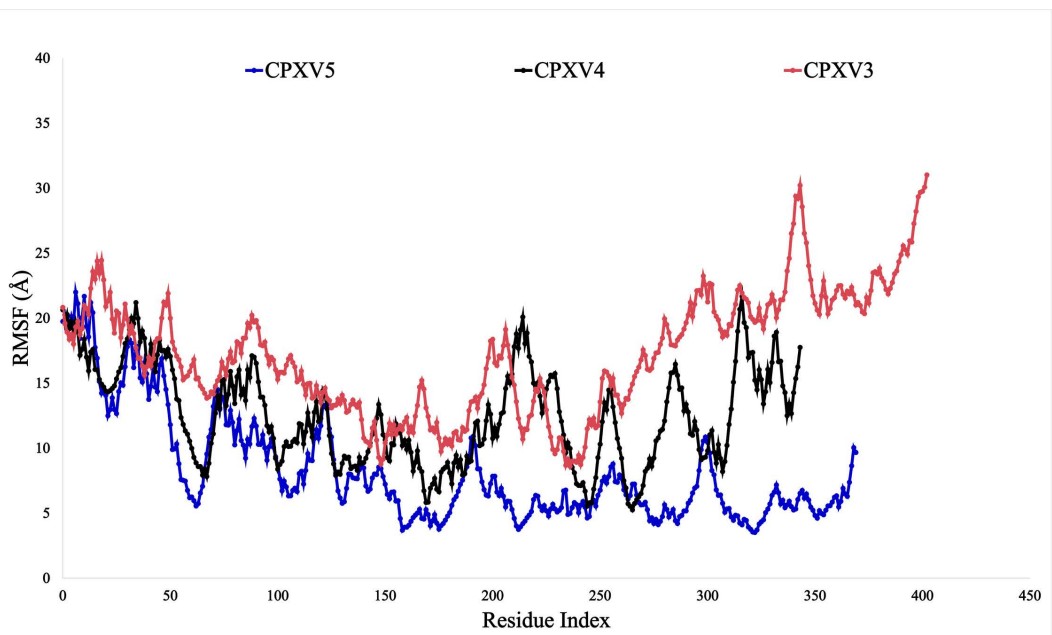

**Fig 8. RMSF analysis from MD simulation.** The RMSF of the top three proteins, red, blue, and black-colored protein A35, protein resolve A22, and scaffold protein.

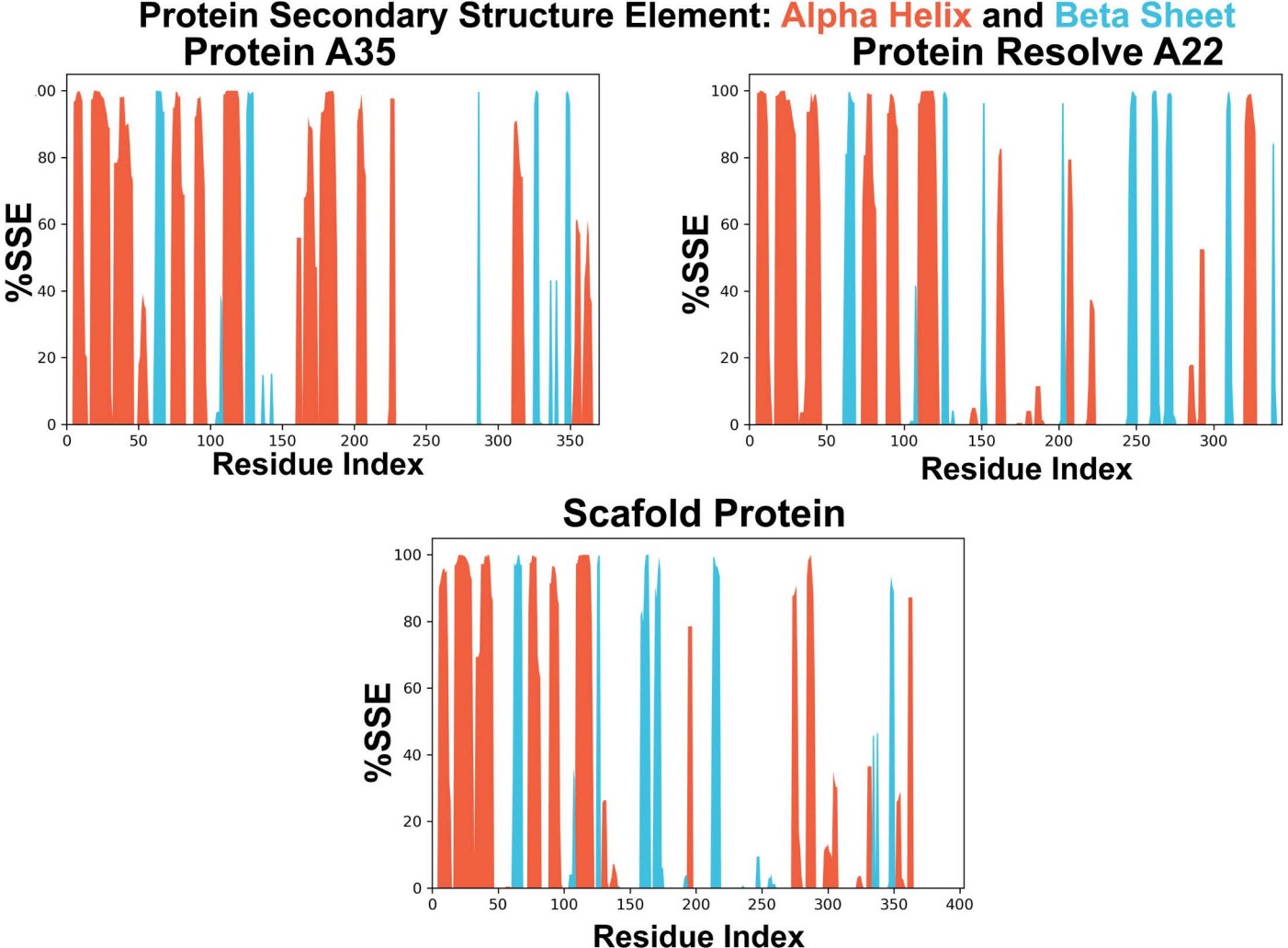

**Fig 9. SSE Analysis from molecular dynamic simulation.** Three proteins, protein scaffold, protein resolve A22, and protein A35, each have different secondary structures. The beta-sheet is colored blue, while the alpha helix is colored red.

**Table 4. Physiochemical features of the vaccine constructions applying ProtParam and JCAT server.**

| Vaccine construct | Number of Amino Acids | Molecular Weight (Daltons) | Antige- nicity Score | Theoretical pI | Aliphatic index | Grand average of hydropathicity (GRAVY) | Instability index | GC content | CAI |
|---|---|---|---|---|---|---|---|---|---|
| Protein L1 | 388 | 41479.17 | 0.5733 | 5.01 | 80.08 | −0.080 | 23.75 stable | 59.05 | 0.95 |
| Protein J5 | 301 | 32720.31 | 0.5592 | 8.73 | 101.26 | −0.061 | 30.99 stable | 47.17 | 0.96 |
| Scaffold protein D13 | 403 | 43943.76 | 0.7315 | 8.85 | 87.87 | −0.086 | 15.37 stable | 46.73 | 0.92 |
| Resolvase A22 | 344 | 37183.35 | 0.6482 | 9.25 | 94.27 | −0.121 | 35.76 stable | 45.93 | 0.98 |
| Protein A35 | 370 | 39469.64 | 0.6673 | 8.44 | 93.08 | −0.002 | 31.61 stable | 59.09 | 0.96 |

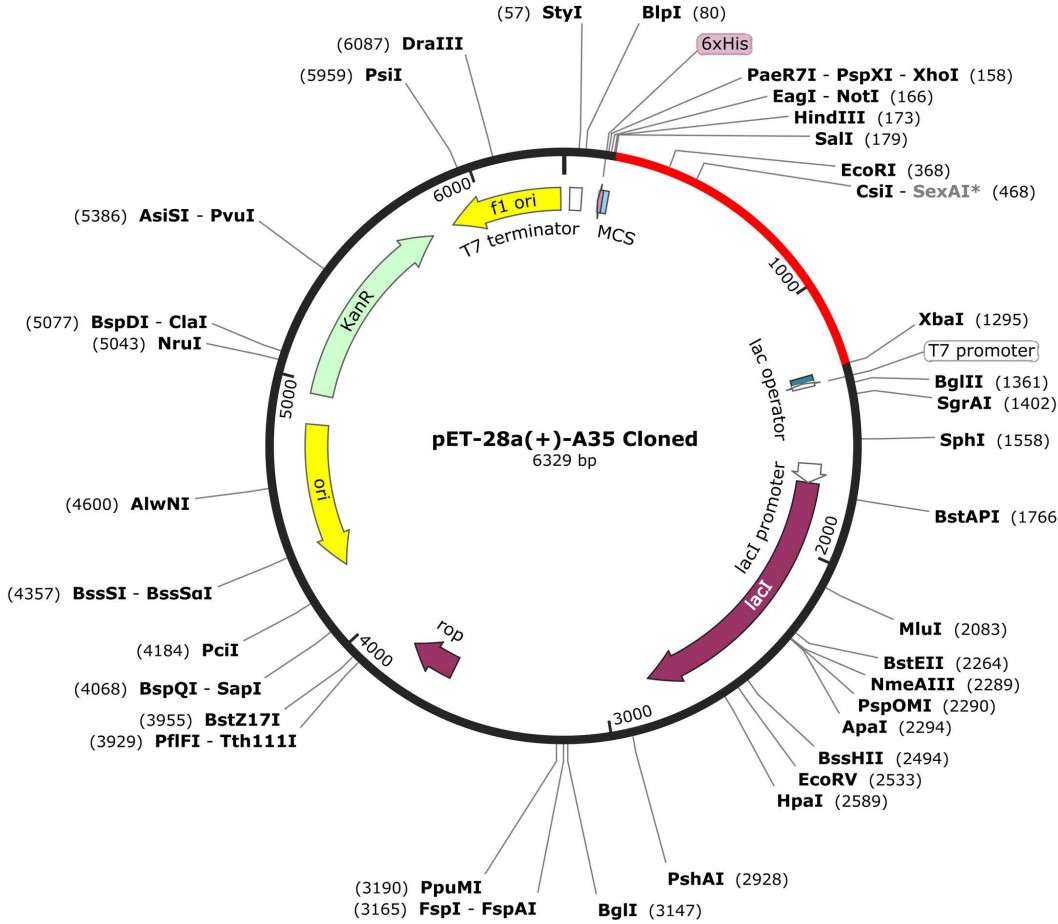

**Fig 10. In silico cloning of the designed vaccine construct into an expression vector.** In silico cloning of the CPXV-V5 vaccine into the pET28a (+) Escherichia coli expression vector.

The list of epitopes identified from each of the nine most potent fragments was further analysed in terms of immunogenicity, which includes antigenicity, allergenicity, cytokine production, and toxicity. Linkers and adjuvants were then used to join specific epitopes to create the sequence. The N-terminal was modified to include an adjuvant, which is 50S ribosomal protein L7/L12, and an EAAAK linker. AAY linkers were then employed to fuse the epitopes [72]. Due to its antibacterial and immunomodulatory qualities, b-defensin is an excellent adjuvant and has been employed in several prior research. These modifications involved incorporating some AAY linkers to ensure that certain epitopes retain functions that are useful once delivered into the host body. The final vaccination was created by combining them with the appropriate linkers: MHC class-I epitopes are connected through AAY linkers, KK linkers for GPGPG linker to the neighbouring MHC class-II epitopes, and B-cell epitopes [73,74].

Based on the presence of the alpha helix, docking, and dynamic experiments were used to justify and validate the vaccination structure for Protein L1, Protein J5, Scaffold protein D13, Resolvase A22, Protein A35. Studies on the physicochemical properties of vaccination were carried out to pave the way for passing experimental exercise of the vaccine and to set viable experiments in vitro and in vivo the standard. The proposed five vaccines depict the fact that epitope vaccination possesses the prospect of provoking high immune responses, and at the same time, the epitope vaccination does not trigger allergy reactions. The results revealed that compared to the other five vaccine constructs, Vaccine CPXV-V5 had more

immunogenicity soluble thermostability and less allergenicity. Allosteric properties of the protein containing its function, additional proteins, ligands, and other changes in its properties can be discovered at the tertiary and secondary levels [75]. After the expected change in the 3D structure, certain three characteristics of vaccine CPXV-V5 improved significantly. According to Ramachandran plot analysis, 92% of residues are situated in the favourable zone, and 0% are in the banner area, which supports the availability of model quality. The overall performance of the present immune simulation was also very encouraging and provided a relatively constant immune response of the cell immunogenicity that occurs normally. The finding demonstrated that memory B-cells are in charge of boosting the immune system's response after several vaccinations. The T-cells that are T-cytotoxic and T-helper cells were also activated by IFN-γ and IL-2, indicating that the vaccination's humoral immunity is strengthened by repeated exposure to the antigens. To ensure the stability of the vaccine candidate, we looked at the construct's ability to identify various MHC class-I, II, and TLRs. These elements are necessary for the activation of immune cells, which sets off the adaptive immune response. The vaccine design (CPXV-V5) has a higher binding affinity of 3C5J (HLA-DRB302:02), according to the molecular docking result. This will cause both an intrinsic and adaptative immunological response to the induced vaccination [76]. In order to assess the dynamic stability of the complex. The molecular dynamic simulation produced a number of pictures that demonstrated how the docked complex stays bound and creates robust molecular interaction with the immune receptor. By joining cystine bridges to the final vaccine structure, cystine engineering improves protein thermostability and facilitates the analysis of the vaccine's genetic makeup. A serological immunoreactive test is one of the methods used to evaluate the advantages of a potential vaccination [77].

In immunological simulation validation, our candidate vaccine exhibited the highest rise in IFN-g, while IL-10 and IL-2 had substantial activity. Increased concentrations of active immunoglobulins, namely IgG and IgM, have also been detected. The different forms of these immunoglobulins and the process of switching between them have been noticed. A vaccine candidate should possess the morphology that allows it to reach out to the host immune system receptors like TLR2 and TLR3 for transportation in the host to be as effective as possible [78]. Besides backing the strong interaction between TLRs and the vaccine in molecular docking and 100 ns MD simulation, they have also shown in the MMGB(PB) SA study that deficient energy was required for this interaction. Small fluctuations were observed as compared to the MD simulations. These findings suggest that the vaccine has to have a strong affinity for immunological receptors and be efficiently delivered throughout the body. An analysis of molecular docking studies showed that the vaccination designs demonstrate a high binding affinity with the active receptor site of the virus. It validates the ability of the vaccination to induce long-term immunogenic reactions. The choice of the best vaccination candidate of high stability, productivity, and fitness to optimally dock, thereby providing better binding free energies, was made from the following docking postures and atoms' interactions. Consequently, CPXV-V5 is deemed to be the structure with the lowest global power of the vaccine construct. It was considered for investigations into immunological and molecular dynamic simulation and later with its two nearest neighbours in the vaccine constructs.

The right host has to synthesize the recombined protein. After carrying out in silico cloning, Reverse transcription, Codon optimization, and RNA secondary structure analysis, it was concluded that the E. Coli K12 system would highly express our planned Vaccine. Last of all, the pET28a (+) plasmid of E.coli was added with the vaccine peptide. The above target vaccine fragment was cut and inserted into the pET28a (+) vector to obtain a Cloned plasmid. The SalI restriction enzyme at the N-terminal and the XbaI enzyme at the C-terminal generated the plasmid's restriction site.

Even if the computational approach for vaccine design versus CCHVF is held to high standards, several shortcomings might still be addressed and further researched in subsequent research. For instance, while the vaccination is immunogenic, additional research is necessary to ascertain the exact extent of immunological defence against the illness. The selection criteria for post-analysis and epitope filtering were stringent, which helped the developed vaccine become a strong contender against the Capripox virus even though the present findings are provisional and need verification by in vitro and in vivo experimentation. In short, the true immunogenicity of the proposed vaccine construct for use in clinical settings needs to be assessed through experimental evaluation.

## Conclusion

The present work used reverse vaccinology and immunoinformatic techniques to identify suitable therapeutic vaccine candidates against CPXV, considering antigenicity, allergenicity, virulence, and toxicity. The chosen proteins were used to generate multi-epitope vaccination constructs by integrating appropriate linkers and adjuvants based on the predicted B and T-cell epitopes. The recommended vaccine design (CPXV-V5) showed a notable affinity for TL4 immune receptor proteins, particularly demonstrated by molecular docking and MD simulation. The expected humoral response, via virus-specific antibodies, may neutralize free viral particles and prevent infection, in contrast to the cellular response, notably cytotoxic T-cell activation, which is likely required for the clearance of infected hosts. Such responses cumulatively reflect significant correlations of defense against CPXV, as evidenced by immunological results in similar poxvirus treatment models. The CPXV-V5 vaccine designs have intriguing practical potential as a preventative strategy against CPXV infections, particularly in susceptible groups. It might also be used as a framework for orthopoxvirus competence and quick vaccine production against new pathogenic threats. Even though these computational outcomes are encouraging and offer a solid basis, in vivo tests and clinical research are still required in suitable animal models to establish the vaccine's immunogenicity, safety, and protective effectiveness.

## Supporting information

**S1 Fig. RMSD plot of the Capripox virus vaccine construct showing structural stability during MD simulation.**
(XLSX)

**S2 Fig. RMSF (Root Mean Square Fluctuation) plot of the Capripox virus vaccine construct showing residue-level flexibility during MD simulation.**
(XLSX)

**S3 Fig. Graphical abstract.**
In this Fig, we exhibited the whole process of the Capripox virus vaccine design.
(TIF)

## Author contributions

**Conceptualization:** Md. Mohaimenul Islam Tareq, Sattyajit Biswas, Farazi Abinash Rahman.

**Data curation:** Md. Mohaimenul Islam Tareq, Sattyajit Biswas, Farazi Abinash Rahman.

**Formal analysis:** Md. Mohaimenul Islam Tareq, Sattyajit Biswas, Farazi Abinash Rahman, Sadia Jannat Tauhida.

**Investigation:** Md. Mohaimenul Islam Tareq, Sattyajit Biswas, Farazi Abinash Rahman, Labib Sharirar Siam, Sadia Jannat Tauhida, Shamim Ahmed, Hasan Jafre Shovon, Mariya Ahmed, Kazi Afrin Jerin.

**Methodology:** Md. Mohaimenul Islam Tareq, Sattyajit Biswas, Farazi Abinash Rahman, Labib Sharirar Siam, Sadia Jannat Tauhida, Hasan Jafre Shovon, Mariya Ahmed.

**Software:** Md. Mohaimenul Islam Tareq, Sattyajit Biswas, Farazi Abinash Rahman, Labib Sharirar Siam, Sadia Jannat Tauhida.

**Supervision:** Md. Nazmul Hasan.

**Validation:** Md. Mohaimenul Islam Tareq, Sattyajit Biswas, Farazi Abinash Rahman.

**Visualization:** Md. Mohaimenul Islam Tareq, Sattyajit Biswas, Farazi Abinash Rahman, Sadia Jannat Tauhida, Shamim Ahmed, Hasan Jafre Shovon, Mariya Ahmed, Kazi Afrin Jerin.

**Writing – original draft:** Md. Mohaimenul Islam Tareq, Sattyajit Biswas, Farazi Abinash Rahman.

**Writing – review & editing:** Md. Mohaimenul Islam Tareq, Sattyajit Biswas, Farazi Abinash Rahman.

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
