## [Decision Letter · Decision Letter 0]

Dear Dr. Hasan,

Thank you for submitting your manuscript to PLOS ONE. After careful consideration, we feel that it has merit but does not fully meet PLOS ONE’s publication criteria as it currently stands. Therefore, we invite you to submit a revised version of the manuscript that addresses the points raised during the review process.

We look forward to receiving your revised manuscript.

Kind regards,

Mohammad Habibur Rahman Molla

Academic Editor

PLOS ONE

Journal Requirements:

3. Please ensure that you refer to Figure 2, 3, 8, 9, 10 and 11 in your text as, if accepted, production will need this reference to link the reader to the figure.

4. Please upload a copy of Figure 1, to which you refer in your text on page 10. If the figure is no longer to be included as part of the submission please remove all reference to it within the text.

5. We note you have included a table to which you do not refer in the text of your manuscript. Please ensure that you refer to Table 1,2 and 4 in your text; if accepted, production will need this reference to link the reader to the Table.

Reviewers' comments:

Reviewer's Responses to Questions

**Comments to the Author**

1. Is the manuscript technically sound, and do the data support the conclusions?

Reviewer #1: Yes

Reviewer #2: Yes

2. Has the statistical analysis been performed appropriately and rigorously?

Reviewer #1: Yes

Reviewer #2: Yes

3. Have the authors made all data underlying the findings in their manuscript fully available?

Reviewer #1: Yes

Reviewer #2: Yes

4. Is the manuscript presented in an intelligible fashion and written in standard English?

Reviewer #1: Yes

Reviewer #2: Yes

Reviewer #1: Major Comments:

1. The abstract mentions three proteins (A35, A22, and Scaffold Protein) as potential vaccine candidates, but it would be helpful to briefly explain why these particular proteins were chosen over others.

2. The use of immunoinformatics is well-described; however, a mention of the specific tools or software used in the analysis could provide more context on the methodology.

3. The abstract mentions MD simulations and immune responses, but adding a sentence on how these simulations directly correlate to predicted efficacy in real-world conditions would be beneficial.

4. The use of a bacterial expression system for the CPXV-V5 vaccine is interesting; clarifying whether the expression was successfully scaled up in vivo or only validated in silico would add clarity to the scope of this finding.

5. The conclusion highlights successful docking and simulation results. A clearer connection between these in silico findings and potential in vivo efficacy could further strengthen the conclusion.

6. The need for further in vivo experiments is clearly stated. It would be beneficial to briefly outline the next steps for in vivo validation, such as animal models or specific experiments to be conducted.

7. The mention of both humoral and cellular immune responses is valuable. It would be helpful to briefly specify how these responses might correlate with protection against CPXV in animal models or humans.

8. The conclusion effectively summarizes the main findings of the study. It might benefit from a more explicit statement about the potential impact of the CPXV-V5 vaccine in real-world applications.

9. The manuscript refers to several data sources but does not mention how the quality and relevance of the data were assessed. It would be useful to include a brief discussion on the selection criteria for these datasets.

10. Some of the figures lack detailed captions explaining the results clearly. Expanding the captions to provide more context or a summary of the findings would help readers understand the significance of each figure.

11. The manuscript discusses the potential for vaccine development, but it could further explore the challenges or limitations of implementing reverse vaccinology and immunoinformatics for Capripox virus. A brief mention of potential hurdles such as variability in viral strains or computational limitations would be beneficial.

Reviewer #2: Dear authors,

This is a study to present the development of potential vaccine against capripox virus. Available datasets were utilized and reverse vaccinology and immunoinformatic techniques.

In general, it is recommended that the acronym needs carefully to go through this manuscript, for example, SPV and SPPV, GPC and GTPV…etc.

All legend of tables and figures are described clearly, and describe in text appropriately.

Lastly, it is also considered the English editing.

**Do you want your identity to be public for this peer review?** For information about this choice, including consent withdrawal, please see our Privacy Policy

Reviewer #1: No

Reviewer #2: No

---

## [Author Response · Author response to Decision Letter 1]

26 May 2025

Reviewer #1: Major Comments:

Comment 1: The abstract mentions three proteins (A35, A22, and Scaffold Protein) as potential vaccine candidates, but it would be helpful to briefly explain why these particular proteins were chosen over others.

Response 1: Thank you for your comment. We selected these three proteins as potential vaccine candidates based on their antigenicity scores, which can be found in Table 4. Additionally, we conducted molecular dynamics simulations to further validate these potential vaccine candidates.

2. The use of immunoinformatics is well-described; however, a mention of the specific tools or software used in the analysis could provide more context on the methodology.

Response 2: Thank you for your valuable feedback. We appreciate your suggestion regarding the inclusion of specific tools used in our immunoinformatics analysis. In response, we have revised the Methods section to include the names and versions of the software and online tools utilized in our study.

3. The abstract mentions MD simulations and immune responses, but adding a sentence on how these simulations directly correlate to predicted efficacy in real-world conditions would be beneficial.

Response 3: I appreciate your thoughtful comment. In response to your recommendation, we have added a statement that explains the relationship between the outcomes of molecular dynamics (MD) simulations and the anticipated effectiveness of the vaccine constructions in practical settings. Particularly, we emphasize how the simulations' results of molecular durability and extended antigenic properties allow validity to the vaccine's probable effectiveness on the immune system when administered in a living organism.

4. The use of a bacterial expression system for the CPXV-V5 vaccine is interesting; clarifying whether the expression was successfully scaled up in vivo or only validated in silico would add clarity to the scope of this finding.

Response 4: I appreciate your wise observation. Computational cloning analysis was used to confirm the CPXV-V5 vaccine's bacterial expression system results in silico by using the SnapGene software. This investigation does not involve any in vivo (experimental) validation because we don’t have this scope. We already mentioned in the abstract and in the methodology section to make it explicit that the gene expression analysis was done computationally.

5. The conclusion highlights successful docking and simulation results. A clearer connection between these in silico findings and potential in vivo efficacy could further strengthen the conclusion.

Response 5:

6. The need for further in vivo experiments is clearly stated. It would be beneficial to briefly outline the next steps for in vivo validation, such as animal models or specific experiments to be conducted.

Response 6: Thank you for your useful suggestion. To provide a quick overview of the future stages, including in vivo validation using mouse models, we have updated the conclusion. These additions have been incorporated into the conclusion section.

7. The mention of both humoral and cellular immune responses is valuable. It would be helpful to briefly specify how these responses might correlate with protection against CPXV in animal models or humans.

Response 7: Thank you for the insightful comment. The conclusion has been changed to describe how the projected humoral and immune responses within cells potentially influence CPXV immunity. Humoral response prevents viral entry by producing neutralizing antibodies, while the cellular response activates cytotoxic T cells to remove infected cells. We have incorporated that clarification in the modified conclusion to emphasize the immunological significance of the CPXV-V5 construct.

8. The conclusion effectively summarizes the main findings of the study. It might benefit from a more explicit statement about the potential impact of the CPXV-V5 vaccine in real-world applications.

Response 8: Consequently, we have updated the conclusion to provide a clearer mention of the possible practical uses of the CPXV-V5 vaccination. Our study has particularly addressed its potential importance in avoiding CPXV cases, improving orthopoxvirus safety, and providing an instance for future vaccine formulations against new pathogenic threats.

9. The manuscript refers to several data sources but does not mention how the quality and relevance of the data were assessed. It would be useful to include a brief discussion on the selection criteria for these datasets.

Response 09: Thank you for your insightful comment. We appreciate the importance of clearly stating the quality and relevance of the data used. In this study, all protein sequences were retrieved from the NCBI Virus database, a well-curated and widely trusted public resource that provides high-quality, annotated viral genomic and proteomic data. To ensure data reliability and relevance, we selected only complete and reviewed protein sequences specific to Capripoxvirus species (Sheeppox virus, Goatpox virus, and Lumpy Skin Disease virus).

These criteria ensured that only biologically significant and accurate data were used in the downstream analyses. A brief explanation of this selection process has been added in the Proteins Retrieval, Re-annotation, and Pan-genome Analysis section to the revised manuscript for clarity.

10. Some of the figures lack detailed captions explaining the results clearly. Expanding the captions to provide more context or a summary of the findings would help readers understand the significance of each figure.

Response 10: Thank you for the observation. Regarding the figure captions, we value your suggestion. We would want to make it clear that each figure in the paper already has a full caption that summarizes the main conclusions and clearly explains the data. Every caption was meticulously crafted to guarantee that it provides adequate background information and aids readers in comprehending the importance of the related figure. To improve clarity even more, we have reviewed all figure captions again and made small changes where needed.

11. The manuscript discusses the potential for vaccine development, but it could further explore the challenges or limitations of implementing reverse vaccinology and immunoinformatics for Capripox virus. A brief mention of potential hurdles such as variability in viral strains or computational limitations would be beneficial.

Response 11: Thank you for your thoughtful comment. We appreciate your insight regarding the need to address the limitations and future directions of the computational vaccine design approach. We would like to point out that this discussion has already been incorporated into the manuscript. Specifically, we have acknowledged the provisional nature of our findings, emphasized the need for in vitro and in vivo validation, and highlighted the importance of further assessing the actual immunological protection conferred by the proposed vaccine construct. This ensures that the manuscript presents a balanced view of both the strengths and limitations of the study.

Reviewer #2:

This is a study to present the development of potential vaccine against capripox virus. Available datasets were utilized and reverse vaccinology and immunoinformatic techniques.

Response: Thank you for your kind summary of our study. We appreciate your recognition of our approach using available datasets, reverse vaccinology, and immunoinformatics techniques for vaccine development against capripox virus.

In general, it is recommended that the acronym needs carefully to go through this manuscript, for example, SPV and SPPV, GPC and GTPV…etc.

Response: Thank you for your suggestion. We have thoroughly reviewed and corrected all acronyms accordingly.

All legend of tables and figures are described clearly, and describe in text appropriately.

Response: We appreciate your feedback. All table and figure legends have been clearly described and appropriately referenced in the text.

Lastly, it is also considered the English editing.

Response: Thank you. The revised manuscript has been edited for English language and clarity.

---

## [Editor Report · Decision Letter 1]

Development of Potential Vaccine Against Capripox Virus Implementing Reverse Vaccinology and Pan-genomic Immunoinformatics

PONE-D-25-12180R1

Dear Dr. Hasan,

We’re pleased to inform you that your manuscript has been judged scientifically suitable for publication and will be formally accepted for publication once it meets all outstanding technical requirements.

Kind regards,

Mohammad Habibur Rahman Molla

Academic Editor

PLOS ONE
---

## [Editor Report · Acceptance letter]

PONE-D-25-12180R1

PLOS ONE

Dear Dr. Hasan,

I'm pleased to inform you that your manuscript has been deemed suitable for publication in PLOS ONE. Congratulations! Your manuscript is now being handed over to our production team.

Kind regards,

on behalf of

Dr. Mohammad Habibur Rahman Molla

Academic Editor

PLOS ONE